# GOTEN: GPU-OUTSOURCING TRUSTED EXECUTION OF NEURAL NETWORK TRAINING AND PREDICTION

## ABSTRACT

Before we saw worldwide collaborative efforts in training machine-learning models or widespread deployments of prediction-as-a-service, we need to devise an efficient privacy-preserving mechanism which guarantees the privacy of all stakeholders (data contributors, model owner, and queriers). Slalom (ICLR '19) preserves privacy only for prediction by leveraging both trusted environment (*e.g.*, Intel SGX) and untrusted GPU. The challenges for enabling private training are explicitly left open – its pre-computation technique does not hide the model weights and fails to support dynamic quantization corresponding to the large changes in weight magnitudes during training. Moreover, it is not a truly outsourcing solution since (offline) pre-computation for a job takes as much time as computing the job locally by SGX, *i.e.*, it only works before all pre-computations are exhausted.

We propose Goten, a privacy-preserving framework supporting both training and prediction. We tackle all the above challenges by proposing a secure outsourcing protocol which 1) supports dynamic quantization, 2) hides the model weight from GPU, and 3) performs better than a pure-SGX solution even if we perform the pre-computation online. Our solution leverages a non-colluding assumption which is often employed by cryptographic solutions aiming for practical efficiency (IEEE SP '13, Usenix Security '17, PoPETs '19). We use three servers, which can be reduced to two if the pre-computation is done offline. Furthermore, we implement our tailor-made memory-aware measures for minimizing the overhead when the SGX memory limit is exceeded (*cf.*, EuroSys '17, Usenix ATC '19). Compared to a pure-SGX solution, our experiments show that Goten can speed up linear-layer computations in VGG up to $40\times$, and overall speed up by $8.64\times$ on VGG11.

## 1 INTRODUCTION

While deep neural networks (DNN) can produce predictive models with unparalleled performance, its training phase requires enormous data as input. A single data owner may not possess enough data to train a good DNN. Multiple data owners, say, financial institutions, may want to collaborate in training DNNs. Yet, they are often expected to protect the privacy of the data *contributors*. This discourages any collaborative training over global-scale data that is otherwise promising (Cheng et al., 2019). Moreover, to perform prediction using a trained model, *queriers* need to submit their own *private* data (*e.g.*, medical history). Meanwhile, the model owners want to protect the confidentiality of the trained model in the prediction phase as well. The exposure of the (parameters of a) model (to queriers or a third-party cloud server) may reveal information about its training data (Fredrikson et al., 2015), deterring the participation of data contributors. Also, the model itself is of high commercial value. These concerns hinder the deployment of prediction as a service.

An increasingly popular approach to ensure privacy is using a trusted execution environment (TEE) (Cheng et al., 2019; Tramèr & Boneh, 2019) and in particular, trusted processors, *e.g.*, Intel Software Guard Extension (SGX). When a data provider sends some private data to a server equipped with SGX, it can initialize an *enclave* to receive the data in a confidential and authenticated way and subsequently operate on them. Even the untrusted server, who physically owns the enclave, cannot read or tamper the data inside the enclave. This paper investigates the following questions: *Can we support DNN training (and prediction) by using SGX and untrusted GPU while still preserving the privacy of all stakeholders? If so, how much speedup do we gain by using GPU?*

## 1.1 Our Baseline Approach: CaffeSCONE

Arnautov et al. (2016) propose SCONE, a secure container mechanism that allows developers to directly run applications in an SGX enclave with almost zero code change[1]. We combine SCONE with Caffe (Jia et al., 2014), an efficient open-source DNN framework, to build our baseline privacy-preserving DNN framework – CaffeSCONE. Beyond demonstrating what one can get by applying a generic solution that uses SGX (SCONE) for training (not supported by Slalom), our CaffeSCONE implementation enables more benchmarking for insight in possible improvements, which are eventually achieved by our main result (hence further optimizing it is not our goal). For one, we show (in Section 4.2) that this baseline approach greatly suffers when the enclave's memory limit is reached. Specifically, it invokes a native *paging* mechanism to swap data in and out, which further requires en/decryption. Also, we found that using more threads and cores cannot improve performance.

## 1.2 Our Proposed Framework: Goten

**Secure Outsourcing to GPU**   By using SGX solely, CaffeSCONE is already orders of magnitude faster than the state-of-the-art cryptographic solutions (SecureML (Mohassel & Zhang, 2017), MiniONN (Liu et al., 2017), Gazelle (Juvekar et al., 2018), and DiNN (Bourse et al., 2018), while only SecureML supports training). Nevertheless, in general, CPU (with or without SGX) is not optimized for costly operations in DNN such as matrix multiplication. Using specialized hardware such as GPU for such computation is a common practice. However, SGX-enclaves cannot directly leverage GPU because its security guarantee is bounded within the CPU and fixed memory. It is unclear how CaffeSCONE (and other works including TensorSCONE, Chiron (Hunt et al., 2018), and MLCapsule (Hanzlik et al., 2018)) can leverage GPU without trusting it (or losing privacy).

The SGX+GPU mode of our framework, which we call Goten, enables an even more efficient approach. To the best of our knowledge, no existing work ever explored this possibility on *privacy-preserving training*. A recent work Slalom (Tramèr & Boneh, 2019) also uses GPU but it only offers prediction privacy. We follow the common practice in the cryptographic privacy-preserving training literature (SecureML, its subsequent work (Wagh et al., 2019), and other prior works (Nikolaenko et al., 2013a;b)) which employ non-colluding servers. Specifically, our framework uses three non-colluding GPU-enabled servers, two of them with a trusted processor. This setup appears to be necessary when the primary goal is to achieve privacy without heavyweight cryptographic tools. In practice, one can employ cloud service providers who are market competitors and value their reputations, or involve a government agency especially in healthcare/financial settings.

**Taking Full Advantage of the Servers**   We choose to exploit the server-aided setting fully and employ one additional server when compared with SecureML. What this server does is to "bootstrap" the *triplets* for *secret sharing* (Beaver, 1991) across the two servers, which SecureML assumes such a bootstrap has been done in advance in an offline phase. Goten thus achieves a higher throughput without worrying that the offline preparation will be "exhausted" when the demand reaches its peak, which is also a hidden problem not addressed by Slalom. It also means Goten provides a "true" outsourcing solution – the time needed for securely outsourcing the job to the untrusted GPU is less than that for computing the job locally by the SGX *plus any time needed for pre-computation*. If desired, one may easily adapt our framework back to the two-server setting. (See Section 2.2.)

**Dynamic Quantization Scheme**   We quantize the neural network parameters to fixed-point number format for efficient cryptographic operations (*cf.*, *static* quantization in Slalom). This process needs to be implemented carefully for the following reasons. First, the many matrix multiplications in neural network may scale up the output values quickly, easily exceeding the numeric limit of the data type. Second, there are functions that map values to a small interval (*e.g.*, $\mathrm{softmax}()$ and $\mathrm{sigmoid}()$) which require high precision. To avoid these potential accuracy problems, we developed a *data-type conversion scheme*, again, for enjoying "the best of both worlds," *i.e.*, the benefit of accurate floating-point operations on trusted processors and efficient fixed-point operations on GPUs. Our experiment (Section 4) confirms that our framework preserves high accuracy.

---

[1]TensorSCONE (Kunkel et al., 2019) employed SCONE with TensorFlow (Abadi et al., 2016a) (a DNN framework like Caffe we used); unfortunately, it is not open source.

**Memory-aware Implementation** A naïve solution of overcoming the memory limit of SGX enclaves to rely on the Linux's paging provided by Intel SGX SDK. However, it imposes much performance overhead ranging from $10\times$ to $1000\times$ comparing to unprotected programs (Arnautov et al., 2016) for exiting the enclave mode and switching back after processing the untrusted memory. Hence, in our framework, we take extra measures to reduce the memory footprints by looking into our specific DNN operations and handle any needed memory swapping by the enclave itself.

## 1.3 TECHNICAL CONTRIBUTIONS

Using both SGX and GPU for privacy-preserving training may sound straightforward, but we stress that we tackled a number of issues. To better understand the obstacles, here we revisit how Slalom performs privacy-preserving *prediction* and why it fails to support training. The core idea of Slalom can be described in simple terms: first apply static quantization on an input $x$ to be protected, then outsource the job of computing $f(x + r)$ to GPU by hiding $x$ with a *blinding factor* $r$ in $\mathbb{Z}_q$ (where $q$ is a large prime). Since it focuses on linear layers, $f$ is linear and hence $f(x + r) = f(x) + f(r)$. When SGX gets back $f(x + r)$, it performs "unblinding" using $f(r)$ and obtains $f(x)$. For such outsourcing to be possible, $f(r)$ should be precomputed. As simple as it may seem, Slalom needs to minimize the following three kinds of overheads – (i) computations over $\mathbb{Z}_q$ performed by the untrusted GPU for the security of the blinding trick, (ii) the communication between TEE and the untrusted GPU, and (iii) loading the precomputed unblinding factor $f(r)$ to TEE. Looking ahead, our outsourcing protocol faces even greater challenges regarding (i) and (ii). Slalom addresses (iii) by assumption – it was done in an offline stage before the TEE needs to process any query. If we just ask the SGX to compute it, computing $f(r)$ is of the same complexity as $f(x)$. Another way is to load them on-spot. It is again subjected to the memory limit and incurs the unwanted communication overhead. More importantly, it is insecure to ask the untrusted environment to compute $f(r)$.

There are five conceptual challenges remain unsolved by Slalom regarding training. 1) Dynamic quantization: Slalom explicitly left it as one of the open challenges. 2) DNN weights are fixed at inference time, but it is not for training. This further complicates the dynamic quantization issue since the weights fluctuate. 3) The pre-computation technique does not apply for training. In more details, the training function is actually parameterized by a publicly-known weight $W$, *i.e.*, $f_W(x)$ multiples $x$ with $W$. Moreover, the weight changes after (a batch of) operations are processed which makes $f_W(r)$ useless for a changing weight $W'$. 4) It is now apparent that Slalom does not protect the model weight $W$, which should be protected in private training (and "more private" prediction). This is also one of the open challenges left explicitly by Slalom. 5) The last one is a challenge unique to our solution in addressing the other challenges. In their usage, TEE and GPU are co-located. However, in our settings, we need to propose an outsourcing solution which is efficient enough even we are subjected to an even higher communication overhead between the servers.

Goten is the first framework that preserve the privacy of not only the prediction queries but *the training data and model parameters* with GPU and trusted environment. Our work achieves the highest efficiency of training and prediction in such privacy setting. This is the also first work which performs extensive experimental investigations of this possibility. Concretely, in our case study on VGG, we can speed up a linear layers up to $40\times$, and improve the performance of VGG11 by $8.64\times$.

## 2 SYSTEM MODEL

There are $n$ mutually untrusted data providers who want to jointly train a DNN using their disjoint training data, but they are not willing to reveal their private data to others. They have already agreed on a specific DNN architecture. The corresponding code for the training algorithm is assumed to be genuine after manual or automated verification (Sinha et al., 2016). After training, a querier can obtain prediction results from the resulting DNN and the results are only revealed to the querier.

### 2.1 CAFFESCONE

Fig. 1a shows the system architecture of CaffeSCONE. The server $S$ initializes an enclave $E$ with the specified training and prediction algorithms. The data providers $C_1, C_2, \ldots$ attest $E$ and verify that it is running the intended algorithms. Then they establish a secure channel with $E$ to send it

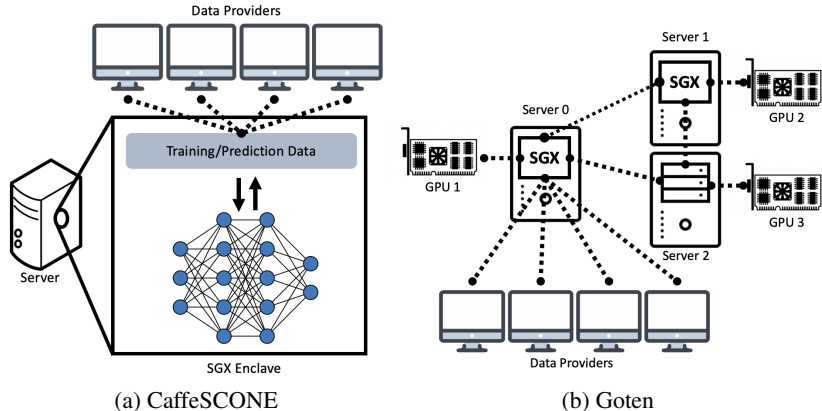

(a) CaffeSCONE          (b) Goten

Figure 1: System Architecture of CaffeSCONE and Goten

their training data. $E$ then trains the DNN with the attested algorithm. Once training is done, $C_i$ sends queries to $E$, which then computes the prediction according to the trained model parameters.

## 2.2 GOTEN

Goten uses GPU to accelerate the computations of the fully-connected and convolutional layers. We introduce two additional non-colluding servers. Fig. 1b illustrates the system architecture.

Servers $S_0, S_1$, and $S_2$ are equipped with GPU and SGX-enabled processor. $S_0$ and $S_1$ initialize $E_0$ and $E_1$ respectively. All the enclaves are attested by the other enclave and data providers, then secure channels are built. $S_0$ and $S_1$ take care of DNN computations. $S_2$ provides multiplication triplets for linear computation (independent of the model parameters or the training/prediction data).

The training and prediction phases are similar to those in (pure-SGX) CaffeSCONE but with two important differences. To avoid cumbersome data transfer between the servers, data providers only send their data to $E_0$, which is then responsible for forwarding to other enclaves. We also design a new outsourcing protocol (from SGX to GPU) that significantly changes the way of matrix multiplication. We leverage the best of SGX (deriving randomness) and GPU (for batch processing). While we still employ a known trick that protects the secret using additive secret sharing, existing designs assume a general scenario and do not consider the characteristics of SGX and GPU.

Our goal is to ensure that an adversary cannot learn anything other than the DNN specification and the data of compromised parties. In particular, the model parameters remain private. Any attacker that observes the communication between all servers cannot compromise privacy. An attacker can compromise any subset of the data providers and at most one of the servers, *i.e.*, two servers cannot collude with each other. We allow the attacker to control all the software (including operating system and hypervisor) of the server, but we assume it cannot launch any hardware attack on SGX. Denial-of-service or side-channel attacks are also out of the scope. See Appendix C for further discussions.

CaffeSCONE further guarantees the correctness of both training and prediction. Goten does not provide it as we present it due to page limitation, but we can resort to the trick used by Slalom.

**Reducing Non-colluding Servers** Our design can be easily modified to use merely 2 servers with some preparation. Looking ahead, the duty of $S_2$ is to produce two random matrices $u, v$, and the product $z = u \cdot v$, and distribute these matrices to $E_0$ and $E_1$. These enclaves can instead prepare $u, v$, and $z$ by themselves, so $S_2$ is no longer needed. Similar tricks are also used by SecureML and MiniONN. Since matrix computation in enclaves is slower than that in GPU, $E_0$ and $E_1$ should pre-compute these matrices before the training/prediction process to prevent stalling the GPU. Additional storage and preparation are required for removing $S_2$.

Moreover, the third server can also be a group of triplet providers which provide triplets in turns. In this case, these providers can amortize the computation requirement so they are not necessarily equipped with expensive GPUs and well-connected with the first two servers.

## 3 THE DESIGN OF GOTEN

### 3.1 HIGH-LEVEL IDEA

Matrix multiplication and convolution occupy $\geq 90\%$ of computation time (see Appendix A.3). It is well known that GPU can speed up the computation of linear transformation and convolution by orders of magnitude. We thus outsource linear operations to GPU, and prevent leaking information to the hosts of (untrusted) GPU via additive secret sharing. Still, CPU needs to convert data of linear layers into the format used by secret sharing, and then convert the result from GPU back into the normal format for non-linear layers. We call these procedures *pre-processing* and *post-processing* of outsourcing linear operations. If they are not handled properly, the processing time could offset the performance gained from GPU. In the following, we introduce our tricks for reducing the run-time of pre/post-processing, and present our modified secret-sharing protocol that improves performance.

Moreover, not only the computation in linear layers but also pre/post-processing suffer from overheads due to paging. We apply *memory-aware measures* to reduce such overhead. The high-level idea is to let the enclave specifies the piece of memory going to use, read and write the memory without triggering Linux's inefficient paging. This approach is also vital for the performance.

### 3.2 GPU-POWERED OPERATIONS VIA OUR OUTSOURCING PROTOCOL

A trivial approach to protect two operands $a$ and $b$ via SGX is to encrypt them to the enclave and ask it to multiply them directly. Yet, it cannot leverage the batch-processing advantage of GPU and is inefficient for large scale computation. We aim to design a protocol that leverages the SGX enclave to secure the unprotected computation environment of GPU, without the enclave performing any expensive decryption beyond the bare minimum, *i.e.*, two decryptions (for the two operands).

We start with the "bare minimum" operations which let the two enclaves $E_0$ and $E_1$ know the secrets $a$ and $b$. The core design principle is to let the enclaves do what they are good for, *i.e.*, generating cryptographic randomness and using them to one-time-pad some values. With the non-colluding assumption (required by the original protocol (Beaver, 1991)), we choose to fully exploit it and introduce one additional server to establish the triplets involved in computing $u \otimes v = z$. The triplets generation can be performed by "the initiating client" offline in existing protocols (Mohassel & Zhang, 2017; Liu et al., 2017), thus, this server can be removed as discussed in Section 2.2.

Fig. 2 describes our protocol for outsourcing linear operation of $c = a \otimes b$. $\otimes$ can be convolution (so $a$ and $b$ are tensors) or matrix multiplication (for matrices $a$ and $b$). Another important usage of enclaves is to store the same seed for deriving the random factors across all the servers. This trick forms a confidential channel between two servers very efficiently without AES or public-key encryption. For example, $S_2$ sends $z$ in the form of $z - \mathrm{Rand}(r_z)$ to $E_0$ and $E_1$ via insecure channels, which can be computed quickly. In other words, all instances of "$\to E_i : var$" in the figure refer to loading the variable(s) $var$ to $E_i$ directly without encryption.

The steps in line 3 of Fig. 2 appear to be working on many more values than the trivial approach of computing $a \otimes b$. Our experiments in Section 4.2 confirms that the performance gain can be as large as $40\times$. Below, we discuss the changes we made over the original triplet-based protocol.

**Parallelizable Pre-Processing without Communication**  Our protocol makes further improvements/refinements over the existing one (in Appendix A.4). Our goal is to compute $a \otimes b$ by operating over $(e, f)$, a masked version $(a, b)$. In the original protocol, the shares $(\langle a \rangle_0, \langle b \rangle_0)$ and $(\langle a \rangle_1, \langle b \rangle_1)$ from the two parties ($S_0$ and $S_1$ here) must be masked independently by the corresponding one-time pads $(\langle u \rangle_0, \langle v \rangle_0)$ and $(\langle u \rangle_1, \langle v \rangle_1)$. After this step, they must interact to produce $e$ and $f$.

In our protocol, both enclaves know $a$ and $b$, so they can use the same seed to derive the same one-time pads $u$ and $v$ (which is in, say, $\mathbb{Z}_q^m$) and obtain $e$ and $f$ without any interaction. This saves half of the pre/post-processing and communication cost, and makes $e$ and $f$ no longer dependent on $\langle a \rangle_i$ and $\langle b \rangle_i$. All the steps in line 3 of Fig. 2 thus can be done in parallel. We then further reduce the run-time of such pre-processing roughly by $3/4$, *i.e.*, it is $1/4$ of the original. Moreover, $E_0$ and $E_1$ no longer need to interact until the last step for result construction, they can then work in parallel.

---

**Secure Outsourcing of Linear Operation $\otimes$ to GPU**

---

$1:$    $S_2 : u \leftarrow \mathrm{Rand}(r_u), v \leftarrow \mathrm{Rand}(r_v), z = u \otimes v, \langle z \rangle_1 \leftarrow z - \mathrm{Rand}(r_z)$

$2:$    $S_2 \rightarrow E_0, E_1 : \langle z \rangle_1$

**for** $i = 0, 1$ in parallel:

$3:$    $E_i : \langle a \rangle_i \leftarrow \mathrm{Gen}_i(a, r_a), \langle b \rangle_i \leftarrow \mathrm{Gen}_i(b, r_b), e = a - \mathrm{Rand}(r_u), f = b - \mathrm{Rand}(r_v),$
      $\langle z \rangle_0 \leftarrow \mathrm{Rand}(r_z), K_{0 \rightarrow 1} \leftarrow \mathrm{Rand}(r_{k_0}), K_{1 \rightarrow 0} \leftarrow \mathrm{Rand}(r_{k_1})$ in parallel;

$4:$    $E_i \rightarrow S_i : \langle a \rangle_i, \langle b \rangle_i, e, f, K_{i \rightarrow 1-i}$

$5:$    $S_i \rightarrow E_i : c_i = \langle a \rangle_i \otimes f + \langle b \rangle_i \otimes e - i \cdot e \otimes f$

$6:$    $S_i \rightarrow E_{1-i} : C_{1-i} = c_i - K_{i \rightarrow 1-i}$

**endfor**

$7:$    $E_0 : c = c_0 + (C_0 + K_{1 \rightarrow 0}) + \langle z \rangle_0 + \langle z \rangle_1$
      $E_1 : c = c_1 + (C_1 + K_{0 \rightarrow 1}) + \langle z \rangle_0 + \langle z \rangle_1$

---

Figure 2: Protocol for Outsourcing Linear Operation $\otimes$

**Reducing Run-time of Share Reconstruction**    Unlike the original standalone protocol where each party only needs to learn a share $\langle c \rangle_i$ of $c$ but not $c = a \otimes b$ itself, it is necessary for our enclaves to know $c$ because they need to perform the succeeding non-linear operations of non-linear layers. (In some existing protocols, $c$ is actually recovered "implicitly" via cryptographic means, say, within a garbled circuit.) A naïve way is to let $S_i$ encrypt their respective shares to the other enclave $E_{1-i}$. Again, we use the common seed to form a secure channel which lets $S_i$ one-time-pad its own share $c_i$ into a ciphertext $C_{1-i}$ for $E_{1-i}$ via the key $K_{i \rightarrow 1-i}$ derived from the seed. In total, we reduce pre/post-processing time by roughly 87.5% and halve the communication cost.

**Performance Gain for Linear Layers**    Our outsourcing protocol, while optimized, still imposes overhead in pre/post-processing and communication between the servers. It is instructive to confirm how much we gain. Beyond the obvious reliance on the relative performance of the GPU, it turns out to be crucially relying on the shapes of the input and weight (specifically, *arithmetic intensity* (cud, 2019)). Appendix D gives the theoretical analysis. Fig. 5a shows convolution gains speed-up as expected when paging overhead is low.

### 3.3   DATA TYPES AND DYNAMIC QUANTIZATION

The triplet trick we used operates over fixed-point numbers in $\mathbb{Z}_q$, while common neural network framework operates over floating-point numbers ("floats"). Therefore, Goten has to accommodate the fixed-point setting so that it can attain superior performance as if using floats.

**The Choice of $\mathbb{Z}_q$**    GPU is slow in modular arithmetic, off-the-shield optimized libraries do not support them. To work on $\mathbb{Z}_q$ integers, we thus put them as floats as Slalom (Tramèr & Boneh, 2019). This leaves us only 53 significant bits plus a sign bit to represent the integers in linear layers (where the rest of $(64 - 53 - 1)$ exponent bits are 0).

To make sure the result of the matrix multiplication or tensor convolution $a \otimes b$ does not overflow, we need $q^2 k < 2^{53}$, where $k$ is the number of columns of matrix $a$ or $k = C_{in} \cdot f_w \cdot fw$ in convolution.

To avoid overflow in $\mathbb{Z}_q$, $q$ should be large; but predicting the value of $k$ beforehand is hard. We thus resort to the heuristics of testing different choices of $q$ over common VGG networks. Based on our experiments, $q = 2^{21} - 9$ is the largest value that does not overflow in almost all ($\approx 100\%$) cases.

**Challenges in Quantization**    To compute $x \otimes_f w$ with floating-point multiplication $\otimes_f$, we need a quantization scheme to convert floats to fixed-point numbers and vice versa for linear layers. We first quantize $x$ and $w$ into $x_Q = Q(x; \theta_x)$ and $w_Q = Q(w; \theta_w)$, where $\theta_x$ and $\theta_w$ are the corresponding quantization parameters. We then use fixed-point multiplication $\otimes_{\mathbb{Z}_q}$ to compute $y_Q = x_Q \otimes_{\mathbb{Z}_q} w_Q$, and derive the result by $y = Q^{-1}(y_Q; \theta_x, \theta_w) \approx x \otimes_f w$.

Slalom only supports prediction. Knowing the model, it knows the value distribution of model parameters. It can then derive the distribution of the input, output, and intermediate values. Picking a *static* scaling parameter that minimizes the error in prediction is thus relatively easy. In Slalom, $Q(\cdot; \theta)$ is always parameterized by $\theta = 2^8$ for all data (inputs and weights) and every computation. In short, static quantization may not pose a big problem in a prediction-only framework.

**Dynamic Quantization for Training**   Slalom clearly states that quantization for training is a challenging problem. For training, the range of gradient of the weight may change, hence the output, and the input of the successive layer. Knowing the value distribution prior to training is hard, so we cannot determine what parameters for $Q$ is good enough to support training.

Beyond what Slalom did, we need *dynamic* quantization for training, meaning that it can adapt the change on the distribution of the model parameters, and hence the intermediate value and gradient. The (de-)quantization process has to be *efficient* since it is part of the pre(/post)-processing of our GPU-powered scheme. An inefficient scheme would reduce or even offset the performance gain.

**Our Choice**   SWALP (Yang et al., 2019) is a training scheme which works in a low-precision setting. The forward and backward computation are performed in low-precision fixed-point, but the weights are stored and updated in floats with high-precision.

Suppose bit is the number of bits available for the low-precision computation, and the default value is $8$. For both the weight and the input, SWALP first finds out the maximum absolute value, and then calculates its exponent in the format of bits, *i.e.*, compute $\mathsf{exp} = \lfloor (\log_2 \circ \max \circ \mathrm{abs})(data) \rfloor$. Then, it scales up all the values by that exponent so that the new maximum values are roughly aligned to $2^{\mathsf{bit}} - 2$, rounds them up stochastically (Gupta et al., 2015), and clips all the value to $[-2^{\mathsf{bit}} - 1, 2^{\mathsf{bit}} - 1 - 1]$, *i.e.*, $data_Q = Q(data, \mathsf{exp}) = \mathsf{clip}(\lfloor data \cdot 2^{-\mathsf{exp} + \mathsf{bit} - 2} \rfloor)$. After the computation, the resulting values are scaled down accordingly, *i.e.*, $y = y_Q \cdot 2^{\mathsf{exp}_x + \mathsf{exp}_w - 2 \cdot \mathsf{bit} + 2}$

Based on the existing SWALP experiment, its accuracy drops by less than $1\%$ when compared to training in full-precision for VGG16, and the operands are only of $8$ bits. Also, finding the maximum absolute value and scaling up and down the values only requires 3 linear scans. The scaling can be fused with other pre/post-processing too. Finally, this scheme matches with our expectation that it is dynamic because it samples the maximum value of the weight and input every iteration. Section 4.2 shows that with this quantization scheme, Goten can train VGG11 to attain high accuracy efficiently.

## 3.4   MEMORY-AWARE MEASURES

When the allocated memory in the enclave exceeds the 128MB limit, it incurs excessive overhead. Our memory-aware mechanism handles most operations in the enclave to mediate this problem.

A naïve solution is Linux's paging, which is provided by Intel SGX SDK. However, native paging is known to be inefficient. As reported in SCONE (Arnautov et al., 2016), memory access can be $10 - 1000\times$ slower compared to plaintext setting. Eleos (Orenbach et al., 2017) explains that triggering SGX native paging would make the CPU core exit the enclave mode, which is time-consuming. The more memory allocated, the more frequent such expensive operations are invoked.

To prevent these expensive operations, our memory-aware measures restrict the memory usage of the computations in SGX to minimize the chance of native paging. When Goten needs to allocate memory more than 128MB, it would directly encrypt the chunk of memory and evict it to the un-trusted zone, which, unlike the native paging, does not leave the enclave mode. When it needs to use memory that is not in the enclave, it loads the chunk of memory into the enclave and decrypts it. Section 4.2 shows that our mechanism speeds up the computation of non-linear layers by $8.72\times$.

For operations inside the enclave, we aim to minimize the memory access across the border between the trusted/untrusted zone. In particular, we fuse together operations that use the same set of memory, and independently handle batches in non-linear layers to prevent excessive use of memory.

Eleos (Orenbach et al., 2017) is also another mechanism for mediating page-fault overhead. It allows the program to handle page-fault without exiting the enclave. CoSMIX (Orenbach et al., 2019) further automates the instrument for this paging-handling mechanism. However, its implementation was released less than a month, so we have not compared or integrated with it.

## 4 EMPIRICAL EVALUATION

For Goten, its SGX part is written in C++ and compiled with Intel SGX SDK 2.5.101.50123, and we use Pytorch 1.2 (pyt, 2019) on Python 3.6.9 to marshal network communication and operation on GPU, which run with CUDA 9.0. The C++ code is compiled by GCC 7.4. Also, we reuse some code of Slalom (Tramèr & Boneh, 2019), including their code of crypgtographicially-secure random number generation and encryption/decryption, and their OS-call-free version of Eigen, a linear-algebra library. All the experiments were conducted for at least 5 times, and we report the average of the results. We uploaded our source code to https://github.com/goten-team/Goten.

### 4.1 SETUP

**SGX's Simulation Mode and Hardware Mode**   Only limited models of Intel CPU are powered by SGX, which can run in the regular *hardware mode* and enjoy the SGX protection. Intel SGX SDK also provides *simulation mode* for testing purpose. Its code compilation is almost the same as hardware mode except that i) the program is not protected by SGX, which is fine for our purpose since the DNN training and prediction algorithms are *publicly known*, and ii) it does not use encryption, which does not affect our experimental timing figures because we handle most of our secret values via one-time pads. In particular, a ciphertext produced by one-time pad is as long as the plaintext it is encrypting, thus, it does not affect the most important overhead – paging.

In term of performance, the largest difference between these two mode is related to paging. When the allocated memory in enclaves exceeds its physical limit, the enclaves in hardware mode may suffer much larger overhead compare to native programs. In simulation mode, the overhead is little. Programs in hardware mode has negligible overhead as long as no paging is triggered. Specifically, according to the experimental results in Privado (Tople et al., 2018), the neural networks which do not trigger page-fault do not have any performance overhead.

**Experiemental Environment for CaffeSCONE and Goten**   We evaluate the performance of CaffeSCONE on a computer (which supports SGX hardware mode) equipped with Intel i7-7700 Kaby Lake Quad-cores 4.3GHz CPU and 16GB RAM, using Ubuntu 18.04. For reproducibility and for the ease of setting up the experiment, we evaluate the performance Goten on 3 Google Cloud VMs. We specify all VMs to equip CPU with Sky Lake, the latest microarchitecture that can be used for Google Cloud's VM. Unfortunately, all CPUs on Google VMs do not support Intel SGX's hardware mode. Also, all these machines are equipped with 32GB RAM and a Nvidia V100 GPU.

**Calibration on Experiment Results**   Given the constraint, our experiments on the environment we used for Goten would underestimate the performance of programs running in SGX simulation mode because the CPUs have lower clock rate and older microarchitecture compared to Intel i7-7700.

To make the comparison between these two frameworks fair, we calibrate Goten's CPU runtime to CaffeSCONE's CPU runtime. We measure the runtime of the non-linear layers in the two afore-mentioned environments. We found that the environment we used for Goten would overestimate the runtime on CaffeSCONE's CPU. Hence, we decide to scale down the runtime of most time-consuming non-linear layers in Goten according to the data collected. The scaling factor for ReLU is $0.96$, for Batchnorm is $0.56$, for Maxpool is $0.85$.

Since the runtime in linear layers is related to the transfer between CPU and GPU and over the network, it is hard to calibrate the runtime of CPU solely. Also, our data showed that the pre/post-processing CPU time is similar across hardware mode and simulation mode. So we do not calibrate the runtime of linear layers. The results in Fig. 4 and Tables 1 and 2 are calibrated by this method.

**Choice of Dataset and Architecture: CIFAR-10 and VGG11**   Both of Goten and CaffeSCONE are evaluated on CIFAR-10, a common dataset for benchmarking the accuracy. We pick a VGG architecture with 11 layers and batch normalization layers because it is a typical DNN that can attain high accuracy on CIFAR-10. Also, it is small enough to fit with (the memory limit of) CaffeSCONE.

### 4.2 PERFORMANCE ON VGG11

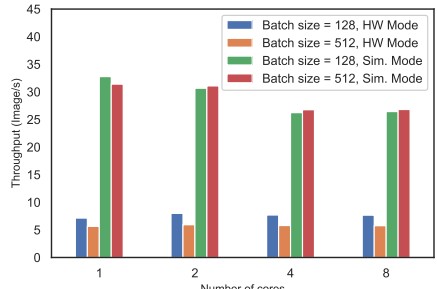

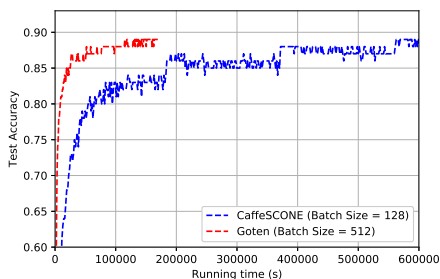

Figure 3: Training Throughput of CaffeSCONE

Figure 4: Accuracy Convergence in VGG11

Table 1: Time Distribution on Linear/Non-linear Layers

|  | Linear Layers | | Non-linear Layers | | Total |
| --- | --- | --- | --- | --- | --- |
|  | Time (ms) | Proportion | Time (ms) | Proportion | Time (ms) |
| CaffeSCONE (BS=128) | 9243 | 57.7% | 6774 | 42.3% | 16017 |
| Goten (BS=512) | 4306 | 58.1% | 3106 | 41.9% | 7412 |
| Speedup | 8.59× | - | 8.72× | - | 8.64× |

**Throughput of CaffeSCONE**   First, we show the training throughput of CaffeSCONE in Fig. 3, by which we emphasize that using more cores on CPU cannot improve the performance of such a pure-SGX approach. Moreover, we benchmark the throughput with batch sizes of $128$ (a common setting in plaintext setting) and $512$ (the setting we adopted for Goten). We confirmed that the former one has better performance for VGG11 in CaffeScone, and thus we adopt it in the later experiments. Note that we adopt batch size to $512$ in Goten because with which Goten has better performance.

**Training Throughput of Goten**   Table 1 illustrates the speedup of Goten compared to CaffeSCONE in the training phase. For the experimental settings, Goten ran with simulation mode on Google VMs and employed memory-aware measures to reduce the overhead of paging. Moreover, we rescale the running time on non-linear layer, which bases on the running time with the real SGX setting, *i.e.*, the hardware mode on the experimental machine equipped with Intel i7-7700.

According to the experimental results on non-linear layers, programs running with the real setting are faster than those on Google VMs. Hence, we believe that linear layers in the real setting are also faster as both kinds of layers have similar operation and (linear) access patterns.

In conclusion, Table 1 shows that Goten outperforms CaffeSCONE by about $8\times$ on linear layers and non-linear layers in VGG11 and by $8.6\times$ on the whole network.

**Convergence on Quantized Neural Networks**   Furthermore, Fig 4 demonstrates how the performance speedup leads to a higher convergence rate. Since the training methods for CaffeSCONE and Goten are different: the former adopts the most common approach, which uses plain single-precision floats, whereas the latter one employs the dynamic quantization scheme SWALP (in Section 3.3). it is natural to wonder whether Goten can attain a higher convergence rate. Our experimental result is confirmative. We record the converge trajectory of both training methods, which was captured in an unprotected setting on GPU, and then rescale the time axis according to the timing from Table 1. The results show that Goten can converge much faster.

To better emphasize our advantage on the convergence rate, Table 2 lists the speedup (at different levels of accuracy), which ranges from $4.93\times$ to $11.78\times$. It shows our quantization scheme does

Table 2: Accuracy vs. Speedup using GPU-powered Scheme

| Accuracy | 0.90 | 0.89 | 0.88 | 0.87 | 0.86 | 0.85 |
| --- | --- | --- | --- | --- | --- | --- |
| Speedup | - | 4.93 | 7.28 | 7.31 | 7.31 | 11.78 |

not have a significant impact on training, and it attains a high accuracy in a shorter time. However, Goten still cannot attain 0.9 accuracy after 200 epochs, while CaffeSCONE can.

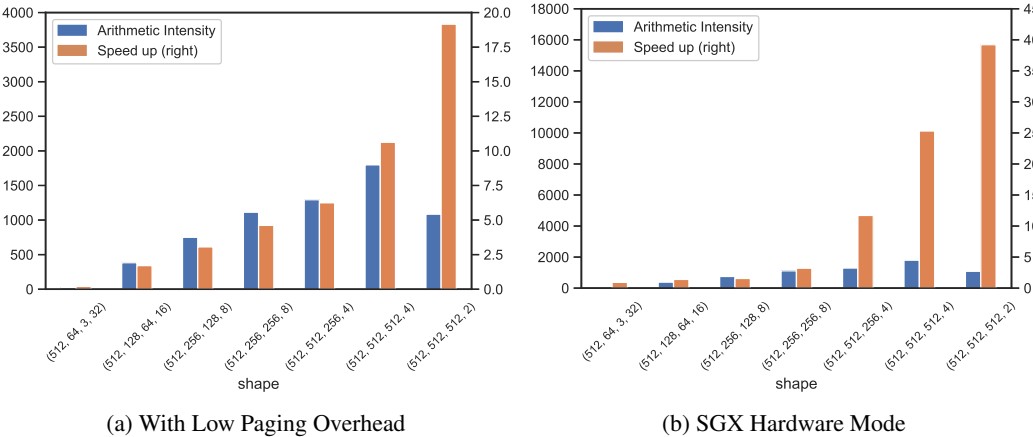

(a) With Low Paging Overhead          (b) SGX Hardware Mode

Figure 5: Speedup vs. Arith. Intensity of GPU-powered Conv. of Shape $(B, C_{out}, C_{in}, I_{hw})$

**Micro-benchmarks: Speedup of Our GPU Outsourcing Protocol**     As our main contribution is the performance speedup on linear layers, we further isolate the performance gain of them. Fig. 5 shows the speedup and arithmetic intensity, which is explained in Appendix D, of each convolution layer presented in VGG with CIFAR-10. The shapes correspond to the batch size, the number of input channels, the number of output channels, the height and width of input images. The filter size of all layers is $3 \times 3$. The results illustrate that Goten are most beneficial to neural networks with high-arithmetic-intensity linear layers.

Fig. 5a shows the result in simulation, where paging overhead is negligible as explained. The result confirms with our analysis in Appendix D: the higher arithmetic intensity a convolution layer has, the higher gains of performance. Furthermore, to have performance gain in our experimental environment, the arithmetic intensity should be at least 250. Also, we notice that the layer with image size $2 \times 2$ actually has a huge performance gain while it has a relatively low arithmetic intensity. We suspect that it is because Caffe cannot efficiently handle inputs with a small image size in the CPU.

Fig. 5b shows the estimated speedup in hardware mode, where paging overhead is significant. The estimation is derived from the same setting of Table 1. The results show a much higher speedup when there are small images and many input channels, and the speedup is not proportional to the architecture intensity. We suspect that the convolution's implementation of Caffe amplifies the paging overhead in the above situation.

## 5    CONCLUSIONS

We proposed a new secure neural network framework using trusted processors. Our framework not only outperforms cryptographic solutions by orders of magnitude, but also resolved the memory limits issues in the existing state-of-the-art trusted processors approach (Ohrimenko et al., 2016). We made privacy-preserving training, prediction, and model-outsourcing for very deep neural networks more deployable in practice by advancing the frontier of the SGX-based machine-learning. For the first time, we can run a very deep neural network, with privacy, but without any memory issue.

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

## A   PRELIMINARIES

### A.1   NEURAL NETWORKS

A neural network gains its predictive power by imitating biological neural networks (Goodfellow et al., 2016). A (feedforward) neural network can be represented by a sequence of transformations.

This paper focuses on supervised learning — every training data is a data point $\mathbf{x}$ associated with a label $\mathbf{y}$, and the neural networks try to learn the relationship between $\mathbf{x}$ and $\mathbf{y}$. Prediction in supervised learning outputs a label of query $\mathbf{x}$.

We refer the computation for prediction by *forward-propagation*. For training, gradient descend is usually employed, where the computation for updating the parameters is called *backward-propagation*.

#### A.1.1   COMMON LAYERS IN NEURAL NETWORKS

Roughly, transformations in a neural network can be divided into two categories: linear transformation and non-linear transformation.[2]

For the linear transformation, we have two kinds of layers.
i) *Fully-connected layer* (a.k.a. *dense layer*) — It just multiplies a weighting matrix to the input (for training or prediction).

ii) *Convolutional layer* — It is similar to the convolution operation except it rotates the kernels by $180$ degrees. The data structure of inputs, outputs, and kernels are *tensors*, which are usually $3$-dimensional or $4$-dimensional.

For non-linear transformation, we have —
i) *Activation layer*, which applies a non-linear function on each element to mimic the impulse activation of biological cells.
ii) *Pooling layer*, which aggregates values in a group after applying a function like $\max()$ or $\mathrm{mean}()$ function.
iii) *Output layer*, which outputs the results in the prediction phase. In the training phase, it computes a loss value measuring the error between the ground truth and the neural network's prediction.

#### A.1.2   COMPUTATIONAL ASPECTS

The linear transformation is the most computationally intensive part (Jia, 2014) when we compute in plaintext. The same applies to the SGX setting. Looking ahead, we will leverage GPU to accelerate the computation of linear layers. Looking ahead, we further outsource the linear transformation to multiple servers by additive secret sharing (Section A.4) to improve efficiency.

In SGX-enclave, the non-linear transformation can be processed in plaintext efficiently. These non-linear transformations are basically aggregating the output from its previous layer and/or applying element-wise operations. A simple but efficient way to handle them is to load the entries from

---

[2]Some weird layers may appear in some architecture but can be easily implemented using the principle we introduced.

the previous layer to the enclave cache memory one-by-one in a deterministic order and output the results once it got enough inputs. In this way, the data remains confidential and the memory access pattern is hidden.

In contrast, without SGX, (cryptographic) solutions either use garbled circuits, resulting in high computation and communication overhead (SecureML (Mohassel & Zhang, 2017), MiniONN (Liu et al., 2017), and Gazelle (Juvekar et al., 2018)), restricted choice of the activation layer and pooling layer (CryptoNet (Gilad-Bachrach et al., 2016)), or dramatic reduction of the size of neural networks (DiNN (Bourse et al., 2018)). As a result, these solutions are not compatible with many well-developed neural network architectures such as AlexNet (Krizhevsky et al., 2017), VGG16/19 (Simonyan & Zisserman, 2015), *etc.*

### A.1.3 Very Deep Convolutional Network (VGG)

This is a family of very deep neural networks with $9 - 19$ layers with parameters (Simonyan & Zisserman, 2015) and has extraordinary performances on object classification. They have convolution layers with similar setting, *e.g.*, all of the convolution has filters of $3 \times 3$ and followed by ReLU and some of them further followed by $2 \times 2$ max-pooling layers. They are commonly used neural networks and hence it is worth to study how to improve the performance of neural networks in privacy-preserving setting.

### A.2 Intel SGX

SGX is the latest Intel hardware-assisted remote secure computing design. Since its seventh generation (Intel, 2017), Intel introduced a set of instructions and hardware design with which an *enclave* can be allocated in the trusted hardware, protecting the privacy and integrity of the data to be processed within it.

### A.2.1 Security Enclaves and Memory Limit

In SGX, enclaves are used as secure containers. When the secure software requests a secure container, an enclave will be loaded with the code and the data specified by the secure software. The enclave will isolate itself from the rest of the computer. Then the data owner can verify the integrity of the enclave by undergoing a standard remote attestation of SGX. Inside an enclave, all the data will be stored in the main memory in an encrypted and authenticated form when the CPU core is not processing them. When some specific data is going to be processed, it will be loaded into memory caches dedicated to a CPU core with SGX protection enabled and then be decrypted.

Although Intel claims that the current SGX supports up to 128MB of memory, at most 90MB is usable according to Shaon et al. (2017).

### A.2.2 Generic Application

The trusted hardware is directly applicable to secure computation. Imagine that a data provider holding some sensitive data wants to perform some secure computation on a remote server. The data provider does not trust the server owner and thus he wants that only the server owner can know the pre-defined output. The trusted processor is an efficient solution satisfying these requirements: data can be processed in plaintext inside the trusted processor but remains unknown and tamper-proof, even to the server owner. Of course, the data owner needs to trust both the software provider and the hardware manufacturer.

### A.3 Graphics Processing Unit

A GPU consists of thousands of cores that can perform similar instructions in parallel. If an algorithm is parallelizable, GPU can increase its computation performance by orders of magnitude.

The most computationally intensive part of neural networks can be transformed into matrix computation, which is well-suited for GPU. Jia (2014) showed that fully-connected layers and convolutional layers occupy over $95\%$ computational time. Abdelfattah et al. (2016) concluded that GPU can speed-up matrix multiplication by $\geq 10\times$ compared to multi-core CPU.

### A.4 Two-Party Computation via Secret Sharing

For two servers $P_0$ and $P_1$ holding private input $a, b \in \mathbb{Z}_q$ respectively, where $q$ is a prime, they can let a third server learn $c = a + b \in \mathbb{Z}_q$ without revealing $a, b$ as follows. $P_0$ chooses a uniformly random $a' \in \mathbb{Z}_q$, then sends $\langle a \rangle_1 = a'$ to $P_1$, and keeps $\langle a \rangle_0 = a - a'$. $P_1$ does a similar job: samples and sends $\langle b \rangle_1 = b'$ to $P_0$, and keeps $\langle b \rangle_0 = b - b'$. No one revealed $a$ or $b$ in this process. Then, $P_0$ computes $\langle c \rangle_0 = \langle a \rangle_0 + \langle b \rangle_0$ and $P_1$ computes $\langle c \rangle_1 = \langle a \rangle_1 + \langle b \rangle_1$. At this point, $P_0$ and $P_1$ both hold (additive) secret shares of $c = a + b$. Any third party with both shares $\{\langle c \rangle_i\}$ can learn $c = \langle c \rangle_0 + \langle c \rangle_1$.

Beaver (1991) generalized the above method to let $P_0$ and $P_1$ compute secret shares of $c = a \cdot b$ as follows. Suppose $P_0$ and $P_1$ have already pre-computed additive secret shares of $u, v$, and $z$ where $u \cdot v = z$. Namely, $P_i$ has $\langle u \rangle_i, \langle v \rangle_i$, and $\langle z \rangle_i$. $P_i$ masks $\langle a \rangle_i, \langle b \rangle_i$ via $\langle e \rangle_i = \langle a \rangle_i - \langle u \rangle_i$ and $\langle f \rangle_i = \langle b \rangle_i - \langle v \rangle_i$. They then exchange $\langle e \rangle_i$ and $\langle f \rangle_i$ to reconstruct $e$ and $f$, which is masking $a$ and $b$ respectively. Finally, with $e$ and $f$, they compute $\langle c \rangle_i = -i(e \cdot f) + f \cdot \langle a \rangle_i + e \cdot \langle b \rangle_i + \langle z \rangle_i$ locally, where $\langle c \rangle_0 + \langle c \rangle_1 = ab$. This technique can be further generalized to matrix addition/multiplication by replacing $\mathbb{Z}_q$ with $\mathbb{Z}_q^{m \times k}$ or $\mathbb{Z}_q^{k \times n}$. Indeed, this technique can applied to any linear operation, including convolution.

Using this protocol as-is requires two rounds of communication (for recovering $(e, f)$) and pre-computation (of shares of $(u, v, z)$). Looking ahead, we will illustrate how to reduce the communication cost and the pre-computation and hence improve the throughput.

In the rest of the paper, we use $\text{Rand}(r_x)$ to denote a function that takes a random seed $r_x$ and outputs a random element $x' \in \mathbb{Z}_q$. Then the (additive) secret share of $x$ held by $P_i$ can be written as $\langle x \rangle_i = \text{Gen}_i(x, r_x) = i \cdot x + (-1)^i \cdot \text{Rand}(r_x)$.

## B Related Work

### B.1 Cryptographic Solutions

Gilad-Bachrach et al. (2016) proposed CryptoNet. It exploits non-linear functions supported by leveled homomorphic encryption (LHE) and parallel computation to improve the efficiency of neural network evaluation. However, it only supports limited activation function ($x^2$ or $\text{sigmoid}(x)$) and pooling function (average pooling). The experiment results of CryptoNet showed that it is roughly $1000\times$ slower than running a similar neural network in plaintext.

Subsequent works (Mohassel & Zhang, 2017; Liu et al., 2017; Juvekar et al., 2018) improve or extend CryptoNet in various dimensions. SecureML (Mohassel & Zhang, 2017) uses two non-colluding servers to support both training and prediction for neural networks, but it is slower than CryptoNet for prediction. MiniONN (Liu et al., 2017) achieves higher prediction accuracy than SecureML for the same network structure. It is also $5\times$ faster than SecureML for small neural networks via the single instruction multiple data (SIMD) batching technique on LHE.

To the best of our knowledge, Gazelle (Juvekar et al., 2018) is the state-of-the-art cryptographic approach in terms of latency. It performs much better than CryptoNet/MiniONN by delicately choosing the HE scheme with optimized parameters to fit the hardware architecture. Gazelle has much lower latency than MiniONN/SecureML as its plaintext space is at most 20 bits. However, it is still unclear whether Gazelle harms the accuracy, which is not stated in their paper (Juvekar et al., 2018).

DiNN (Bourse et al., 2018) follows an approach similar to CryptoNet's. It does not require user interaction during the evaluation. To the best of authors' knowledge, it is the state-of-the-art pure-HE-based approach. Yet, as stressed in the DiNN paper (Bourse et al., 2018), they aim to show that a pure-HE approach is possible and can outperform CryptoNet, at the cost of lower accuracy.

In general, all frameworks mentioned above use expensive cryptographic primitives, such as LHE, garbled circuits, and oblivious transfer, during (training and) prediction, resulting in huge data and computation overheads. Also, using these primitives usually requires multiple rounds of communication between different parties.

As a final remark, there are cryptographic solutions that protect the privacy of (mostly the prediction phase of) other machine learning algorithms. A non-exhaustive list includes decision trees or random

forests (Tai et al., 2017; Wu et al., 2016; Bost et al., 2015), logistic regression (Slavkovic et al., 2007; Bost et al., 2015), support vector machine (Vaidya et al., 2008; Yu et al., 2006), collaborative filtering (Tang & Wang, 2017; Zhao & Chow, 2015), and $k$-means clustering (Bunn & Ostrovsky, 2007; Jagannathan & Wright, 2005). They are conceivably less powerful than a deep neural network.

### B.2 TRUSTED EXECUTION ENVIRONMENT

**Memory Limit**  Ohrimenko et al. (2016) proposed data-oblivious machine learning algorithms using SGX for training and prediction. Their work also defends against some potential side-channel attacks using oblivious operations. However, their algorithms cannot handle any layer of size that exceeds the amount of usable memory (90MB) in an enclave.

The memory limit has been a huge drawback of SGX. Different efforts have been devoted to resolving this issue. Shaon et al. (2017) proposed SGX-BigMatrix. It supports operations on matrices which size exceed 90MB, but still have very high overhead comparing to optimized libraries for unprotected matrices. Linux's SGX supports memory oversubscription for enclaves, but it introduces overhead for the incurred paging, which is reported widely (Weichbrodt et al., 2018; Chakrabarti, 2017; Harnik & Tsfadia, 2017; Brenner et al., 2016; Arnautov et al., 2016). Intel official forum even reported examples of $10\times$ to $350\times$ overheads (Feng, 2017). Moreover, based on our experiments, Linux's paging introduces up to $24\times$ runtime on matrix multiplications.

Orenbach et al. (2017) proposed Eleos, a memory handling mechanism to reduce performance overhead due to SGX's memory page fault. Its main idea is to prevent exiting enclaves when page fault happens because it is an expensive instruction. The experimental results showed that it can reduce the paging overhead by $5\times$. And its successive work CoSMiX (Orenbach et al., 2019) shows that the paging overhead can be further reduced to $1.3 - 2.4\times$. We assume Goten and CaffeSCONE employ this memory handling mechanism to handle paging, and we simulate the performance that does not affected by paging by using simulation mode form Intel SGX SDK.

**TEE-based Approaches and TEE+GPU-based Approaches**  A few proposals rely on TEE (Hunt et al., 2018; Tople et al., 2018) or TEE and GPU  (Volos et al., 2018; Tramèr & Boneh, 2019).

Chiron (Hunt et al., 2018) assumes the data provider shards training data into $n$ pieces for $n$ enclaves, such that each shard fits in enclave memory. The authors left the policy for managing insufficient enclave memory as future work. Most importantly, Chiron requires new SGX features that are not available yet. Volos et al. (2018) proposed Graviton, an architecture for supporting TEE on GPU with the help of SGX, which supports neural network computation in particular, with near-native performance compared to untrusted GPU. However, they assume that an attacker cannot physically steal information from the GPU cores, which is questionable because GPU cores, unlike SGX, are not designed for trusted operation and their security is not well examined.

Bahmani et al. (2017) proposed an SGX-based framework for general-purpose secure multi-party computation. On one hand, our work can be viewed as realizing a specific functionality under their framework at a conceptual level. On the other hand, the general-purpose treatment does not take into account the characteristics of neural network computations. More importantly, we provided important technical contribution and we use untrusted GPU to further accelerate computations. Kunkel et al. (2019) proposed TensorSCONE to port another popular DNN framework TensorFlow to SCONE. Our baseline approach is similar to this framework, but we provide our implementation to public for benchmarking.

Privado (Tople et al., 2018) allows a model owner to outsource privacy-preserving DNN inference to an SGX-enabled cloud server. It guarantees that even a powerful cloud who sees the SGX enclave memory access pattern does not learn model parameters or the user query. Compare with our solution, Privado does not handle training phase, nor does it leverage untrusted hardware like GPU for acceleration.

Tramèr & Boneh (2019) recently proposed Slalom for verifiable and private inference using a trusted enclave which also outsources some computation to a GPU. Their approach heavily relies on the assumption that the server knows the model's parameters. It is thus not applicable to privacy-preserving training.

### B.3 DIFFERENTIAL PRIVACY

Another line of research focuses on achieving differential privacy (Dwork, 2006; Dwork et al., 2006a;b). Abadi et al. (2016b) propose a differentially private stochastic gradient descent algorithm for deep learning. Shokri & Shmatikov (2015) propose collaborative learning, in which data owners jointly train a deep neural network by exchanging differentially private gradients through a parameter server instead of directly sharing local training data. Although Shokri & Shmatikov (2015) makes it hard to tell whether a specific record exists in the victim's private training set, it does not prevent an adversary from learning macro-feature of the training set. Phong et al. (2018) showed that the parameter server in Shokri & Shmatikov (2015) can extract information about the training set, and proposed to use additive HE to eliminate the leakage during training.

## C  SECURITY ANALYSIS

### C.1  PROTECTION SCOPE

From the perspective of the querier, no one else can learn the prediction query and the corresponding result. For the model, the most valuable information includes the parameter of the neural network (*e.g.*, weights and bias of convolutional filters and fully-connected layer), the accuracy according to the training dataset, and the intermediate results. These explicit parameters of the model would not be known by any data-provider and server (with protection against side-channel attacks described shortly afterwards).

We aim for a practical framework instead of a perfectly leakage-free solution. Following the literature (Juvekar et al., 2018; Mohassel & Zhang, 2017; Liu et al., 2017; Gilad-Bachrach et al., 2016), we do not protect the hyper-parameters such as the learning rate, the number of layers, the size of each layer *etc*. These could be inferred by the querier by timing the interaction with the server or by the server from the memory access pattern. One may hide these by adding dummy storage and computation, which is ought to be inefficient.

Side-channel leakage is also out of our protection scope. Specifically, the access pattern in cacheline may reveal information about the data (Ohrimenko et al., 2016). In our case, max-pooling layers and the $\mathrm{argmax}$ function in the output layer would be exploitable for their branching depending on the intermediate results. Yet, the existing defence (Ohrimenko et al., 2016) can be easily employed by changing the assembly code of $\max()$ in the enclave, and the computation overhead is less than $2\%$ (Ohrimenko et al., 2016).

Model extraction attacks (Tramèr et al., 2016; Fredrikson et al., 2015) can be launched in a blackbox environment, namely, the attacker knows nothing about the model parameters and its architecture but can query the model, whereby he/she duplicates the functionality of the model. We can easily employ two effective mitigations. First, the training data providers can limit the query rate or set up a query quota by consensus. Second, we only return the labels of evaluation results, instead of the confidence values (the values of the output vector) since it is the main attribute being abused by the attackers (Tramèr et al., 2016).

### C.2  REALIZING THE NON-COLLUDING ASSUMPTION

To perform training, companies may join forces and have the motivation to dedicate a better network line between them. A similar argument also applies to the setting in which one of the parties is generally trusted would not collude with others, say, the government. To provide machine learning as a service in the application context of electronic healthcare (e.g., precision medicine), the Department of Health and Human Services (or alike) can take the effort.

Technically, to ensure at least one server remains secure, we resort to the standard practice from the security community on how to fortify a selected machine. We got many possible ways, including but not limited to, i) using different hardware and software configuration from the other server (so a vulnerability in one platform will not lead to an easy compromise of both machines), ii) placing it within the network-level security parameter behind the DMZ, and iii) limited access to the machine instead of public-facing. Even when a machine is compromised by an insider, we can further enforce

access control, which at least holds the entity with special access-permissions accountable when the access-rights are abused, assuming the security of the audit log of the access control system.

## C.3 Operations inside Enclaves

With the use of SGX, the security guarantee is easy to see. In our construction, the data provided by data providers are either stored in the enclave or sealed on server storage. When data is stored inside the enclave, by the security guarantee of SGX, no other party is able to gain any information. When data is stored outside the enclave, we seal the data by an authenticated encryption (AES-GCM) (Costan & Devadas, 2016), which protects the confidentiality and integrity of a sealed block. We also authenticate the meta-data, in particular, the identity and the number of executions of the block, which disallows arbitrary manipulation of the input data by mix-and-match.

Apart from storage, we also perform execution over the data. In our framework, all executions are data-independent — the executions of neural networks have no branching dependent on the data or models parameters. We analysis our implementation to be data-oblivious using PinTool (Luk et al., 2005), a tool for analysis execution trace, to make sure the trace is the same given model parameter, training data, and prediction queries. The execution view observed by other parties can thus be simulated by without the actual data.

**Data-oblivious Operations**   The host of an enclave can observe the memory access pattern, even in L2 cache level (Brasser et al., 2017). Hence, we need to ensure algorithms running in enclaves are data-oblivious, meaning that the trace of executed cpu instruction should be the same even given different input data.

Functions involves branching, *e.g.* max, min, may arouse concern on data-oblivious because some optimization of compilers may skip the write instruction if the computed value is equal the original value. For example, the write in $y = \mathsf{max}(y, 0)$ may be skipped if $y$ is indeed large than 0.

Fortunately, we can always use vectorization techniques to avoid such situations. With vectorization techniques, *e.g.*, SSE and AVX, the vectorized read and write instructions will not be skipped since they are atomic and hence no branch depending the data value. Even better, such vectorization techniques are usually automatically employed by common compilers, *e.g.* GCC, with proper flags, *e.g.*, -march=native. All we need to do is manually inspecting compiled assemble code or using trace analysis tools, *e.g.* PinTool (Luk et al., 2005), for automatic verification.

## C.4 Outsourcing to GPUs

The only cryptographic primitive we used in the outsourcing protocol is additive secret sharing, which is commonly used in the non-colluding server setting (Wang et al., 2014; Mohassel & Zhang, 2017) for privacy-preserving machine learning. It is also not uncommon in the bigger context of secure multi-party computation (Hohenberger & Lysyanskaya, 2005; Chow et al., 2009; Demmler et al., 2015). Its confidentiality holds in the strong information-theoretic sense against any adversary without enough shares. This fits with the non-colluding server setting well.

Here, we prove that our modified triplet multiplication is secure, namely, none of the server $S_0$, $S_1$, and $S_2$ can gain any information of the contents of $a$, $b$, or $c = a \otimes c$ (the servers can learn their dimensions). Due to the non-colluding assumption, we only need to prove that the knowledge of each individual server can be reduced to their counterpart the original protocol.

For $S_2$, it knows $u$ and $v$, which are random tensors/matrices and contain no information about $a$, $b$, or $c$. Also, $z = u \otimes v$ derived from $u$ and $v$ contains no extra information.

Speaking at high-level, the extra knowledge of $S_0$ and $S_1$ leaks no meaningful information because it is all one-time padding.

Comparing the original protocol described in Section A.4 with our protocol described in Fig. 2. in our protocol, $S_0$ has extra knowledge $\langle z \rangle_1$, $c_1 + K_{1 \to 0}$, and $K_{0 \to 1}$. Now, we apply the game-hopping technique to prove that our scheme is reducible to the original protocol. Firstly, since $S_0$ does not know $\langle z \rangle_0$ in our protocol, we can replace $\langle z \rangle_1$ by $\langle z \rangle_0$. Then, since $S_0$ also does not $K_{1 \to 0}$, $c_i + K_{1 \to 0}$ can also be replaced by a random matrix/tensor. Likewise, $K_{0 \to 1}$ is just another

```
net = nn.Sequential(
  nn.Conv2d(3, 64, 3, padding=1),
  nn.BatchNorm2d(64), nn.relu(),
  nn.MaxPool2d(2, 2),
  nn.Conv2d(64, 128, 3, padding=1),
  nn.BatchNorm2d(128), nn.relu(),
  nn.MaxPool2d(2, 2),
  nn.Conv2d(128, 256, 3, padding=1),
  nn.BatchNorm2d(256), nn.relu(),
  nn.Conv2d(256, 256, 3, padding=1),
  nn.BatchNorm2d(256), nn.relu(),
  nn.MaxPool2d(2, 2),
  nn.Conv2d(256, 512, 3, padding=1),
  nn.BatchNorm2d(512), nn.relu(),
  nn.Conv2d(512, 512, 3, padding=1),
  nn.BatchNorm2d(512), nn.relu(),
  nn.MaxPool2d(2, 2),
  nn.Conv2d(512, 512, 3, padding=1),
  nn.BatchNorm2d(512), nn.relu(),
  nn.Conv2d(512, 512, 3, padding=1),
  nn.BatchNorm2d(512), nn.relu(),
  nn.MaxPool2d(2, 2),
  nn.Linear(512, 512),
  nn.BatchNorm1d(512), nn.relu(),
  nn.Linear(512, 512),
  nn.BatchNorm1d(512), nn.relu(),
  nn.Linear(512, 10)
)
```

Figure 6: The Architecture of VGG11

random matrix/tensor so it can be replaced trivially. Now, $S_0$ has the view of $S_0$ in the original protocol plus two random matrices/tensors.

Likewise, $S_1$ has extra knowledge of $c_0 + K_{0 \to 1}$ and $K_{1 \to 0}$. Applying the same principles for analyzing $S_0$, it can be reduced to $S_1$ in the original protocol.

## D  ANALYSIS FOR PERFORMANCE GAIN FOR LINEAR LAYERS

We first analyze the case of fully-connected layers. Assume $x \in \mathbb{Z}_q^{m \times k}$ is the input, $w \in \mathbb{Z}_q^{k \times n}$ is the weight, and $y \in \mathbb{Z}_q^{m \times n}$ is the output, We found that we should maximize $\min(m, k, n)$. Since $m$, the batch size, is usually small compared to $k$ and $n$, it is better to be large.

We should minimize the run-time ratio of our GPU-powered matrix multiplication scheme to the vanilla CPU scheme. The forward computation in fully-connected layer is $x \otimes w = y$. The run-time of our GPU-powered scheme is $t_{\text{pre-proc}} \cdot (m \cdot k + k \cdot n) + (t_{\text{post-proc}} + t_{\text{comm}}) \cdot (m \cdot n) + t_{\text{gpu-op}} \cdot (m \cdot k \cdot n)$.

The backward computation computes $dx = dy \otimes w$ and $dw = dy^T \otimes x$, where $dx, dw$, and $dy$ are the gradient of $x, w$, and $y$ respectively, and they are of the same size as their counterparts. Similar to the analysis above, the total run-time of both forward and backward computations is $t_{\text{gpu-scheme}} = (2 \cdot t_{\text{pre-proc}} + t_{\text{post-proc}} + t_{\text{comm}}) \cdot (m \cdot k + k \cdot n + m \cdot n) + 3 \cdot t_{\text{gpu-op}} \cdot (m \cdot k \cdot n)$, while the run-time of the vanilla CPU scheme is $t_{\text{cpu-scheme}} = 3 \cdot t_{\text{cpu-op}} \cdot (m \cdot k \cdot n)$. We denote $t_{\text{extra}} = 2 \cdot t_{\text{pre-proc}} + t_{\text{post-proc}} + t_{\text{comm}}$.

Finally, the run-time ratio of these two schemes is

$$\frac{t_{\text{gpu-scheme}}}{t_{\text{cpu-scheme}}} = \frac{t_{\text{extra}}}{t_{\text{cpu-op}}} \cdot (\frac{1}{m} + \frac{1}{n} + \frac{1}{k}) + \frac{t_{\text{gpu-op}}}{t_{\text{cpu-op}}}.$$

The last term matches with the intuition that GPU governs the performance gain. The pre/post-processing and communication time also play an important role if $1/m + 1/n + 1/k$ is large. Note that the inverse of $1/m + 1/n + 1/k$ is also known as the *arithmetic intensity* (cud, 2019).

The analysis on convolution layers follows the same principle but is more involved. If we assume the image size of input and output are the same, we can have a similar result as fully-connected layers by replacing $m, k$, and $n$ to $C_{out} \cdot f_h \cdot f_w$, $C_{in} \cdot f_w \cdot f_h$, and $B \cdot I_h \cdot I_w$ respectively.

