# OpenReview forum: "Goten: GPU-Outsourcing Trusted Execution of Neural Network Training and Prediction"
_ICLR.cc/2020/Conference — Reject_

### Official Review · AnonReviewer1 · 2019-10-16
**Official Blind Review #1**

**Rating:** 1

**Review:**


Summary
========
This paper proposes a framework for privacy-preserving training of neural networks, by leveraging trusted execution environments and untrusted GPU accelerators.
The system builds heavily on the prior Slalom system, and uses standard MPC techniques (three non-colluding servers, multiplication triplets) to extend Slalom's inference-only protocol to privacy-preserving training.
This is a valuable and hard to reach goal. Unfortunately, the paper's evaluation fails to deliver on its strong promises, by ignoring the high network communication between the non-colluding servers.
Specifically, all experiments were conducted with three servers co-located in a public cloud's LAN. In this setting, it is hard to argue that non-collusion is a valid security assumption as the cloud provider controls all servers (alternatively, if the cloud provider is trusted, then there is no need for any trusted execution or cryptography). If the same experiments were conducted on a WAN, the communication costs would alleviate any savings in computation time.

For these reasons, I lean strongly towards rejection of this paper.

Detailed comments
=================
Extending the ideas in Slalom to support privacy-preserving training is a good research question, and Tramer and Boneh had discussed some of the challenges and limitations towards this in their original paper.
Getting rid of the pre-processing stage for blinding factors by leveraging non-colluding servers is a well-known trick from the MPC literature, but it does not seem easily applicable here.
The problem is that the servers need to communicate an amount of data proportional to the size of each internal layer of the network, for each forward and backward pass. If the servers communicate over a standard WAN, the communication time will be much too high to be competitive with the CaffeScone baseline.
In a LAN, as in this paper's experiments, the network latency is low enough for the network costs to be dominated by computation. But this begs the question of whether servers running in a same LAN (e.g., hosted by a single cloud provider) can really be considered non-colluding. In the considered setup, the cloud provider (Google in this case), could just observe the communication between all servers, thereby breaking privacy.

Another security issue with proposed scheme is the lack of computation integrity. This corresponds to the so-called "honest-but-curious" threat model which often appears in the MPC literature, and this should be acknowledged and motivated.

On the experimental side, the considered baseline, CaffeScone, seems pretty weak. In particular, any optimizations that the authors implement for Goten (e.g., better paging) should also be added to their baseline for a fair comparison.
The numbers in Figure 3 show that the baseline could be optimized a lot further.  A gap between hardware/simulation modes of ~6x seems indicative of sub-optimal paging. Even the single-core, simulation mode throughput numbers seem low for CIFAR10.

The experimental setup is quite confusing. Running the baseline and Goten in different environments (e.g., different CPUs) and then re-normalizing throughputs is somewhat convoluted and prone to mistakes. Why not run all experiments on the same setup?
Similarly, converting between results in SGX's hardware and simulation modes is also not very indicative. The authors note (p. 8) that in SGX's simulation mode "code compilation is almost the same as hardware mode except that the program is not protected by SGX, which is fine for our purpose since the DNN training and prediction algorithms are publicly known". This is fundamentally incorrect!
SGX's simulation mode provides absolutely no security guarantees. It simply compiles the code using the SGX libraries and ensures that the enclaved code performs no untrusted operations, but it does not provide any hardware protections whatsoever. In particular, code running in simulation mode will not be affected by the overhead of SGX's paging, as the memory is never encrypted.
As a result, performance results in simulation mode are usually not indicative of performance in hardware mode. Trying to convert runtimes from simulation mode to hardware mode by comparing times of specific layers is also prone to many approximation errors.

Finally, I had some trouble understanding the way in which Goten quantization works. Section 3.3. mentions that values are treated as floats, but then mentions the use of 53 bits of precision. Did you mean double-precision floats here? But then, aren't modern GPU optimized mainly for single-precision float operations? Section 3.3. also says that the quantization ensures that there are nearly no overflows. What happens when an overflow occurs? I guess that because of the randomized blinding, a single overflow would result in a completely random output. How do you deal with this during training?

Minor
=====
- Typo in abstract: Slaom -> Slalom
- I don't understand the purpose of footnote 3 in Appendix B.2. First, the bibliographic entry for (Volos et al. 2018) explicitly says that the paper was published in OSDI 2018, a top-tier peer-reviewed conference. Regardless, claiming a date for a first unpublished draft of your paper is a little unusual and somewhat meaningless. I'm sure Volos et al. had a draft of their paper ready in late 2017 or even earlier if they submitted to OSDI in XXX 2018. If you want to timestamp your paper, post in to arXiv or elsewhere online.

**Experience Assessment:**

I have published one or two papers in this area.

**Review Assessment: Checking Correctness Of Derivations And Theory:**

N/A

**Review Assessment: Checking Correctness Of Experiments:**

I carefully checked the experiments.

**Review Assessment: Thoroughness In Paper Reading:**

I read the paper thoroughly.

---

### Official Review · AnonReviewer3 · 2019-10-17
**Official Blind Review #3**

**Rating:** 6

**Review:**

The paper builds a privacy-preserving training framework within a Trusted Execution Environment (TEE) such as Intel SGX. The work is heavily inspired from Slalom, which does privacy-preserving inference in TEEs. The main drawbacks of Slalom when extending to training are (1) weight quantization needs to be dynamics as they change during training, and (2) pre-processing step of Slalom to compare u = f(r) isn't effective as the weights change, and running this within TEE is no better than running the full DNN within TEE. In addition, Goten also makes the weights private as opposed to Slalom. Overall, this is a very important contribution towards privacy preserving training and the paper takes a strong practical and implementation-focused approach by considering issues arising due to memory limitations in TEE and the performance implications of default Linux paging.

The paper comes up with a novel outsourcing protocol with two non-colluding servers for offloading linear operations in a fully privacy-preserving way and does detailed analysis of the performance implications. Similar to a lot of other methods for training with quantization, the weights are stored and updated in floats while the computation is performed using quantized values. The experimental results suggest a strong improvement over the CaffeSCONE baseline. One drawback with experiments is the lack of comparison with Slalom for inference if Goten is assumed to be a framework for both training and prediction in a privacy-preserving way.

Another downside of the paper is that a few sections could be improved with their explanation, and there is quite a bit of redundancy in going over the downsides of Slalom and why it can't be used for secure training. For instance,
- Section 1.1: "Our results (referring to Section 4.2) show that CaffeSCONE’s performance greatly suffer from the enclave’s memory limit as it needs an inefficient mechanism to handle excessive use of memory not affordable by the enclave". Here, it's not clear which mechanism is inefficient. Are we talking about mechanisms in CaffeSCONE for reducing memory usage while training and if so, are they somehow inefficient? Or does it mean to imply that we can't train a DNN fully within an enclave due to memory limits?
-  Last paragraph of section 2.2 is unclear. "CaffeSCONE guarantees the correctness of both training and prediction. Goten does not provide it as we present it due to page limitation, but we can resort to the trick used by Slalom". What does the last sentence mean? Does Goten guarantee correctness during training and prediction or not? And what trick from Slalom are we referring to? The blinding trick used for privacy or the Freivalds' algorithm used for correctness?

Overall a strong contribution with supporting experimental results, but the certain parts need further explanation or rewriting for higher rating.

Pros:
- An important contribution in the direction of fully private DNN training and inference within a TEE. Draws inspirations from Slalom and mainly addresses the challenges left to extend the approach to training.
- Motivation and reasons for why Slalom can't be used for training is very well laid out.
- In addition to input and output activations, Goten also preserves the privacy of the weights.
- Good baseline for comparison using CaffeSCONE.
- Implementation factors considered and analyzed such as tricks as using SGX-aware paging instead of naive Linux paging.
- Strong experiments and benchmarks

Cons:
- Some sections are not explained well and unclear as mentioned earlier.
- How does the inference performance of Goten compare to Slalom given the same privacy and correctness guarantees? This isn't clear from the experiments section.

Minor comments:
- "Slalom" is mis-spelt in line 4 of the abstract.
- There appear to be typos and grammatical errors at many places in the paper. Further proof-reading might be helpful.

**Experience Assessment:**

I do not know much about this area.

**Review Assessment: Checking Correctness Of Derivations And Theory:**

I assessed the sensibility of the derivations and theory.

**Review Assessment: Checking Correctness Of Experiments:**

I assessed the sensibility of the experiments.

**Review Assessment: Thoroughness In Paper Reading:**

I read the paper at least twice and used my best judgement in assessing the paper.

---

### Official Review · AnonReviewer2 · 2019-10-26
**Official Blind Review #2**

**Rating:** 1

**Review:**

The paper proposes a method for privacy-preserving training and evaluation of DNNs. The method is based on a combination of hardware support from a trusted execution enclave (Intel SGX) and an algorithm for offloading intensive computation to unsecure GPU devices and communicating with the trusted environment without losing security guarantees during communication.  Compared to related work on a similar system (Slalom), the proposed system enables secure training in addition to inference.  The approach is based on the use of additive secret sharing to relegate chunks of computation to independent GPU servers.

The evaluation presents experiments that report timings of the proposed system against a baseline. Throughput (images/second) improvements of 8-9x are reported, but the contrast point is unclear (it appears to be CaffeSCONE, but there are several unclear points, summarized below). In addition, a set of results reporting a speed up ratio against attained accuracy of a trained VGG11 network, and a set of results reporting speed up against arithmetic intensity of the workload are given.

I lean towards rejection of this draft, as it has several weaknesses:
- The connection between the evaluation (which mostly focuses on the speed benefits) and the claimed contributions is tenuous at best. This issue is further compounded by clarity issues in the experiments and their description
- The empirical results are unclear due to differences between simulation of SGX capability vs hardware support of SGX capability. It is not clear what part of the results is influenced significantly by this disparity, and more importantly whether all the comparisons are done in an equal footing (for example the reported results comparing CaffeSCONE with Goten are performed in two different regimes). As a byproduct, there is a confusing "scaling factor" described by the authors that is applied to the timings.
- A brief mention is made of the fact that the proposed system does not in fact provide correctness guarantees (unlike CaffeSCONE), but this is dismissed by reference to utilizing the same trick used by Slalom.  However, this trick is not described or motivated.
- The writing in the current draft is of relatively low quality, significantly impacting the readability of the paper and making it hard to understand the contributions and whether they are backed by the presented results.


**Experience Assessment:**

I do not know much about this area.

**Review Assessment: Checking Correctness Of Derivations And Theory:**

I assessed the sensibility of the derivations and theory.

**Review Assessment: Checking Correctness Of Experiments:**

I assessed the sensibility of the experiments.

**Review Assessment: Thoroughness In Paper Reading:**

I read the paper at least twice and used my best judgement in assessing the paper.

---

### Author Response · Authors · 2019-11-15
**Responses to the Official Reviews**

Meta comment.
We thank all the reviewers for the detailed review, which allows us to have concrete revision plan. We are glad that our technical contributions are rightfully observed and appreciated by reviewer 3. Below, we clarify our technical contributions, in particular, how our approach is innovative or at least new, that we can supplement the "fundamentally incorrect" statement (because we focused two aspects but missed one) but the missing part does not affect our experiment. We also further explain our experimental settings and lay out our plan for some more experiments. (Thanks for the inputs!)

No matter whether this is accepted or not eventually, we would appreciate further comments (if possible/applicable) on how well we responded to each issue, and would definitely appreciate and felt encouraged if the score can be at least slightly adjusted (of course, when deemed appropriate).

For sure, we will revise the paper according to the reviews and the responses below.

R1/R3. [Technical Contributions] The system builds heavily on the prior Slalom system.

1. We share the same general idea of SGX+GPU, but we deviated by just considering one-level lower already. We use 2 SGXes. Our tailor-made outsourcing protocol in Section 3.2 leverages the best of SGX (deriving randomness) and GPU (for batch processing). Existing use of the "well-known trick" just consider outsourcing in general and does not take into account the unique characteristics of SGX and GPU.

i. Our protocol thus achieves *real* outsourcing (in terms of total computation time saved). This is different from Slalom's approach, in which the pre-computation time is as much as if it were computed online locally without outsourcing.

(We briefly reiterate Section 1.3 and what is explained by Reviewer 3: Slalom precomputes f(r), outsources the computation of f(x + r), and gets back f(x) by f(x + r) - f(r). The precomputation of f(r) is as slow as computing the real problem instance f(x).)

ii. Further, to better pinpoint a benefit of our new design, even if we stick with the same deployment assumption as Slalom, i.e., we do the offline precomputation online, the whole protocol is faster than running by the SGX alone. In this sense, saying we used "well-known primitive" is a bit more accurate, but we achieved more.

iii. In the literal sense, it did use "three non-colluding servers, multiplication triplets," like some other existing works. From a bird-eye view, it is still "some form" of "outsourcing" and secure multi-party computation, but we use new design principles (e.g., minimizing the use of encryption to minimal, correlated randomness). We stress that the specific goals we aim to achieve, and hence, the technical details of the seemingly similar conceptual building blocks are different.

2. After all, both works process the neural network. We understand that, from a high-level point of view, they must share a somewhat similar template conceptually to deal with different layers of the neural network. We stress that, in terms of the contribution of the new technical design, we specifically address all the open problems left by Slalom, which hindered them from also supporting training.

Specifically, our design works under several conflicting constraints to support training for the first time, while Slalom failed to resolve (with an explicit discussion on the challenges, so they are not something some "easy" extensions can solve). For one, Slalom's approach inherently leaks the (dynamically changing) weights. An overview is given in the second paragraph of Section 1.3, and details are discussed in Sections 3.3 and 3.4.

To conclude, we think it may be a bit oversimplifying to say our system relies heavily on the prior *system* and we merely use "well-known trick," at least for the facts that i) existing such protocols were not designed for outsourcing from SGX to GPU but just treating them as "generic processors" and ii) we tackled all the challenges left behind by Slalom.

---

> ### Author Response · Authors · 2019-11-15
> **2/n**
>
> R1. [Technical Contributions / Experiment] Ignoring the high network communication.
>
> Our experiment (despite being in the LAN setting) did not ignore the communication overhead among the servers. We took a simplistic view focusing on computation and accuracy. In the revision, we will provide experiments to shed light on how fast is "enough,"  i.e., to estimate the effect of network speed of various WAN settings.
>
> It is our premise to rely on communicating servers to achieve something which was not possible with the pure cryptographic approach, pure SGX approach, or "simple" SGX+GPU approach. In retrospect, Slalom achieves better performance than pure cryptographic approaches, by assuming the existence of a trusted processor (even though security researchers have demonstrated some form of side-channel attacks). In a sense, it further motivates research works (that have been) trying to make the assumption needed by Slalom more realistic.
>
> In our case, we solved what Slalom failed to solve, i.e., privacy-preserving training. Further, it is efficient (and accurate) with a fast enough connection between the servers. Our suggestion is thus to use our system when the premise (again, we will quantify in our experiments) is satisfied.
>
> We stress that we are not solving the problem by merely relying on such an assumption. Many challenges remain, as discussed in the paper and our responses (and some of them are mentioned by the reviews as well). For one, our outsourcing protocol minimizes (sequential) communication needed in the original protocol (see "Parallelizable Pre-Processing without Communication" in Section 3.2).
>
> Please see below for further discussions on our experiment setting and how can we realize our assumptions placed on the servers in practice.
>
> R1. [Technical Contributions] In [LAN] setting, it is hard to argue that non-collusion is a valid security assumption as the cloud provider controls all servers
>
> We believe that non-colluding servers do not *have to* be geographically located far apart. Instead, the crucial point is whether they are really *owned* by the same entity. Moreover, as the network bandwidth is increasing continuously, the communication overhead may not be a bottleneck in certain situations.
>
> Even if one indeed place both servers in the same LAN, there can be other justifications for the security. Let us justify with a real-world example. For user authentication of Facebook, they also maintain an internal server that is "more secure" than the public-facing server. E.g., see "an adversary that compromises the web server and the password hashes it stores must still mount an online attack against the PRF service to compromise accounts,"  as described  by https://eprint.iacr.org/2015/644.pdf. Simply put, security there is also "strengthened" for relying on the "PRF secret key" of another server.
>
> For sure, when both servers are compromised, there is no security. It then comes to the classical discussion from Security 101 on how to fortify a selected machine. We got many possible ways, including but not limited to, i) using different hardware and software configuration from the other server (so a vulnerability in one platform will not lead to an easy compromise of both machines), ii) placing it within the network-level security parameter behind the DMZ, iii) limited access to the machine instead of public-facing, etc. We add this discussion to Appendix C.2.
>
> Let us elaborate more on how non-colluding servers assumption are usually realized in existing works (which is briefly mentioned at the end of "Secure Outsourcing to GPU" in Section 1.2). To perform training, companies may join forces and have the motivation to dedicate a better network line between them. A similar argument also applies to the setting in which one of the parties is generally trusted would not collude with others, say, the government. To provide machine learning as a service in the application context of electronic healthcare (e.g., precision medicine), the Department of Health & Human Services (or alike) can take the effort. The better network line is the cost we need to pay to support practical enough privacy-preserving machine-learning training (and inference), something that was not known to be possible (with a comparable level of efficiency and accuracy) before our work.

---

> > ### Author Response · Authors · 2019-11-15
> > **3/n**
> >
> > R1. [Technical Contributions] In the considered setup, the cloud provider (Google in this case), could just observe the communication between all servers, thereby breaking privacy.
> >
> > Strictly speaking, observing the communication alone is not enough to break privacy, but one needs to further look into internal memory. Of course, when the cloud provider controls all the servers, it can as well look into internal memory. This comes to a fine-grained treatment of the concept of "trust" beyond a boolean yes/no discussion. Thanks for raising this concern.
> >
> > In response, we will make this threat model clear. In practice, abstracting some technical details, access control system can also help in terms of security. Assuming an insider attacker does not have the super-user privilege, and the audit log function of the access control system remains secure, it can then hold the entity with special access-permissions accountable when the access-rights are abused. We add this discussion to Appendix C.2.
> >
> > Again, we took a simple treatment with our experiment, but this does not totally nullify the security of our system. As mentioned above, we will also enrich our experiment along this dimension.
> >
> > R1/R2. [Technical Contributions] Lack of computation integrity (a.k.a. correctness guarantee)
> >
> > It is not our aim to provide integrity specifically, which we will acknowledge. Freivald's algorithm (IFIP Congress 1977) used by Slalom provides verifiability for an interactive outsourcing protocol of linear layers. Freivald's technique allows detection with sound error according to the space of the underlying finite field. This is a classical generic technique, which we can use as Slalom since we overate over finite field elements as well. We are happy to discuss more in the appendix.
> >
> > Our work focuses on privacy, yet, it does not mean we achieve that at the cost of integrity. In our case, privacy and integrity are orthogonal security goals and not inherently conflicting with each other.
> >
> > (In the "secure outsourcing" literature from the cryptography/security community, many major works, for example, searchable encryption, have been studied extensively only in the "privacy only" setting. There are also generic transformations that equip integrity to a searchable encryption scheme that does not ensure integrity.)
> >
> > In practice, the servers may not have high incentive to jeopardize the training as they invested much cost to establish a system and gather the data to support training (which is different from the federated learning scenario or distributed learning scenario). We will also add this "motivation/justification."

---

> > > ### Author Response · Authors · 2019-11-15
> > > **4/n**
> > >
> > > R2. [Experiment] The connection between the evaluation (which mostly focuses on the speed benefits) and the claimed contributions is tenuous.
> > >
> > > (Section 4.2) First of all, we show the training throughput of CaffeSCONE in Fig. 3, by which we emphasize that using more cores on CPU cannot improve the performance of such a pure-SGX approach. Moreover, we benchmark the throughput with batch sizes of 128 (a common setting in plaintext setting) and 512 (the setting we adopted for Goten). We confirmed that the former one has better performance for VGG11 in CaffeScone, and thus we adopt it in the latter experiments. Note that we adopt batch size to 512 in Goten because with which Goten has better performance.
> > >
> > > Table 1 illustrates the speed up of Goten compared to CaffeSCONE in training phrase.
> > >
> > > We first explain the experimental settings. Goten ran with simulation mode on Google VMs and employed memory-aware measures to reduce the overhead of paging. Moreover, we rescale the running time on non-linear layer, which bases on the running time with the real SGX setting, i.e., the hardware mode on the experimental machine equipped with Intel i7-7700.
> > >
> > > According to the experimental results on non-linear layers, program running with the real setting are faster than those on Google VMs. Hence, we believe that linear layers in the real setting are also faster as both kinds of layers have similar operation and (linear) access patterns.
> > >
> > > According to Table 1, in VGG11, Goten outperforms CaffeSCONE by about 8 times on linear layers and non-linear layers.
> > >
> > > Furthermore, Table 2 demonstrates how the performance speed up leads to higher convergence rate. The training methods for CaffeSCONE and Goten are different.
> > > The former adopts the most common approach, which uses plain single-precision floats. The latter one adopt employs the dynamic quantization scheme SWALP, explained in Section 3.3. Hence, it is natural to wonder whether Goten can attain a higher convergence rate. Our experimental result is confirmative. We record the converge trajectory of both training methods, which was captured in an unprotected setting on GPU, and then rescale the time axis according to the timing from Table 1. The results show that Goten can converge much faster.
> > >
> > > To better emphasize our advantage on the convergence rate, Table 2 lists the speed up (at different levels of accuracy). We can attain 0.88 accuracy by about 7 times faster.
> > >
> > > As our main contribution is the performance speed up on linear layers, we further isolate the performance gain of them and show it in Fig. 5. Moreover, because in Section XX we propose that the performance gain is portional to the arithmetic intensity, we hope to confirm it by more fine-grained benchmark. Hence, we record the performance gain with respect to the shapes of linear layers, which determines the arithmetic intensity.
> > >
> > > Fig. 5 (a) is derived from the experiments that both CaffeScone and Goten ran on simulation mode and on Google VMs, and Goten does employ the chunked-operations. As we mentioned above, in simulation mode programs run as fast as those in single-thread normal environment, meaning that the paging overhead of SGX does not affect the performance of both of them.
> > >
> > > Below we also provide a partial list of the figures and tables in our paper accompanied by the different settings.
> > > [Figure 3: Training Throughput of CaffeSCONE]:
> > > CaffeSCONE runs in HW mode
> > >
> > > [Figure 4: Accuracy Convergence in VGG11], [Table 2: Attaining accuracy using GPU-powered Scheme]. [Table 2: Attaining accuracy using GPU-powered Scheme]:
> > > CaffeSCONE in HW mode. Goten in mixed mode (i.e., linear layers in sim mode and non-linear layers are rescaled to hw mode)
> > >
> > > [Figure 5 (a) With Low Paging Overhead] CaffeSCONE: Sim mode. Goten: Sim mode (without memory-aware measures)
> > >
> > > [Figure 5 (b) SGX Hardware Mode] CaffeSCONE: HW mode. Goten: Sim mode (with memory-aware measures)

---

> > > > ### Author Response · Authors · 2019-11-15
> > > > **5/n**
> > > >
> > > > R1. [Experiment] Optimizations should also be applied to CaffeScone.
> > > >
> > > > First, we provide CaffeSCONE as a baseline approach if one applies a generic solution (SCONE) for making use of SGX for training (not supported by Slalom). Further optimizing it is *not* our goal (for one, it does not use GPU at all). It requires significant engineering effort to implement all our optimization tricks (those without using GPU) to CaffeSCONE. We believe further optimizing it has its own impact as a system research work.
> > > >
> > > > (Until now, no results have applied any optimization on paging to any SGX-protected DNN framework -- a side-evidence that such an optimized framework is not as trivial as one might think, despite the potential impact in the system research community.)
> > > >
> > > > More importantly, our experiments for comparing CaffeSCONE and Goten is done with *low* paging overhead because these experiments are done in the simulation mode (where programs can run as fast as if they are executed in single-thread normal execution environments). In other words, our results are *not* derived from an experiment which gives Goten unfair advantage. Yet, our experiments showed that Goten can still outperform CaffeSCONE by 2.5x to 18x in different linear layers. (Also see Figure 5(a).)
> > > >
> > > > Hence, even in an ideal case where paging overhead is no longer a problem (either by applying our optimization techniques on both frameworks; or by using a non-existent, future version of SGX), Goten can still significantly outperform pure-SGX solutions, specifically, for privacy-preserving training.
> > > >
> > > > R1. [Experiment] Running the baseline [CaffeSCONE] and Goten in different CPUs and re-normalizing throughputs is convoluted and prone to mistakes.
> > > >
> > > > The experiment setting is disadvantageous to us because CaffeScone ran in the CPU with a higher clock rate (4.2GHz of Intel i7-7700 vs. 2.0GHz of Google VM's), meaning our evaluation overestimates the running time of Goten. Moreover, we only re-normalize the non-linear layers, and thus this evaluation method does not affect the timing on the linear layers, whose performance improvement is our main contribution.
> > > >
> > > > That said, we felt sorry that we employed different CPUs since we do not have easy access to production-scale hardware resources (GPU) with SGX on the same platform.
> > > >
> > > > R1. [Technical Contribution/Experiment] "Fundamentally incorrect"
> > > >
> > > > The said statement focused on the code compilation and the protection of the algorithms; it is true that it does not use encryption, which we are glad to fix this "incomplete" statement (in Section 4.1).
> > > >
> > > > That said, "simulation mode will not be affected by the overhead of SGX's paging, as the memory is never encrypted" does not apply here since we use simple exclusive-or (or one-time pad with pseudorandom sequence) which introduce no space overhead. Specifically, no matter if it is encrypted or not (i.e., no matter which mode we consider), such padding does not elongate the data to be processed. Hence the paging aspect is also not affected. (Also, see the discussion above and below.)

---

> > > > > ### Author Response · Authors · 2019-11-15
> > > > > **6/6**
> > > > >
> > > > > R1/R2. [Experiment] Simulation mode.
> > > > >
> > > > > Programs in hardware (HW) mode has negligible overhead as long as no paging is triggered. Specifically, according to the experimental results in Privado [Tople et al., 2018], the neural networks which do not trigger page-fault do not have any performance overhead.
> > > > >
> > > > > [Tople et al., 2018] Tople, S., Grover, K., Shinde, S., Bhagwan, R., & Ramjee, R. (2018). Privado: Practical and secure DNN inference. arXiv preprint arXiv:1810.00602.
> > > > >
> > > > > Moreover, based on our experimental data of non-linear layers, namely, the data for computing scaling factors, the performance of programs in simulation mode is similar to that in HW mode when no page-fault is triggered. Indeed, our SGX-aware chunked operations can totally prevent page-fault in enclaves, and thus the performance overhead due to HW mode is negligible in our case. Furthermore, the values of scaling factors support our claim. The scaling factors are the ratio of 1) the running time of a kind of non-linear layers running in HW mode on a machine equipped with SGX to 2) that in simulation mode on a Google Cloud VM. According to our experiment results, all these values are less than 1, meaning that the running time of the simulation mode on Google Cloud VM overestimates the running time of the full-fledged system.
> > > > >
> > > > > That said, we plan to perform experiments for upper-bounding Goten’s running time. To address the concern on the simulation mode, we will run our experiments (of linear layers) on a machine equipped with the SGX module and a weak GPU. Then, we benchmark as a baseline timing the running time in a fine-grained manner, where we record the running time on the CPU, GPU, and the transfer time between CPU and GPU and the networks. To provide a more accurate estimation for the GPU part, we will record its running time and use it to replace the GPU running time in the baseline timing. This evaluation ensures that the running time on the CPU and SGX is authentic. However, such fine-grained timing degrades the measured perform (compared with the true performance) because we synchronize different computing units before any of them starts transferring data or computation. This does not fully utilize the computational resources and the network.
> > > > >
> > > > >
> > > > > R1. [Technical Contributions] Quantization.
> > > > > It is true that most modern GPU are optimized for single-precision float operation, but single-precision floats are not enough for our secret sharing scheme since it only has 23 bits of significand precision for integer operations. Specifically, to achieve both high performance and high accuracy, 23 bits is not enough because integers with 23 bits will easily overflow after multiplication and a bunch of additions, while modulo operations are slow in GPU. Therefore, a sensible choice is to perform computation with double-precision floats, which have 53 bit of significand precision. We use a modern GPU which is also optimized for double-precision float operation.
> > > > >
> > > > > A high-level idea that we do not run into the overflow problem is that we have a very careful treatment of the available bits (as explained above and Section 3.3).
> > > > >
> > > > > R3. How does the inference performance of Goten compare to Slalom given the same privacy and correctness guarantees? This isn't clear from the experiments section.
> > > > >
> > > > > As argued above (R1/R3), Slalom needs to pre-compute f(r) for computing online f(x). We achieve higher throughput for inference as ours is a real outsourcing solution which the pre-computation is not as expensive as the online computation for the real problem instance.
> > > > >
> > > > > A primary motivation of Goten is to provide a more comprehensive privacy coverage which was not known to be possible (by Slalom) before, namely, to further achieve data privacy (confidentiality) for the model's parameters. For such protection in training (and inference), Goten does introduce overhead (just like all the subtleties we discussed above). Hence, we found it natural that Slalom has better performance on *inference*, because its design inherently cannot support privacy for model parameter, and it is tailor-made for its weaker privacy guarantee confined to only inference. Simply put, again, it fails to support privacy-preserving training.

---

### Comment · AnonReviewer1 · 2019-11-15
**Response to author comments**

I thank the authors for the detailed rebuttal.
Nevertheless, I think that many misconceptions remain, which I describe below:

* On the LAN vs WAN setting:
While one issue in a WAN is bandwidth, the much bigger problem is *latency*. After every layer, the server hosting the SGX enclave has to send data to the server hosting the second GPU, and wait for the GPU's reply before moving on to the next layer. In a WAN, the roundtrip latency will be in the 5-150 millisecond range depending on how geographically close the servers are. Now, multiply this by the number of layers. Running the entire computation inside SGX will always be faster.

Your rebuttal mentions the need of a good network line. Paying more money will indeed get you more bandwidth, but latency is fundamentally limited by the speed of light.

* On non-collusion in a LAN:
The real-world example you give with Facebook's password protections doesn't really relate to your case. Here, Facebook is trying to protect against an outsider adversary. But Facebook obviously has access to the contents of both servers, so this doesn't protect the data from Facebook.
The challenge in Goten is to make sure that no single entity can see the data processed by the two GPUs. If these two GPUs both belong to the same company (e.g., Facebook), then you need to trust that company for privacy. But if you do trust this company for privacy, you can just have them run the entire computation in the clear.

* On access control:
You mention that one way to ensure non-collusion within an organization is to enforce access-control of the servers. Again, this doesn't actually provide any security from the organization itself, which is in charge of the access control. But even so, at this point you could also just run the whole computation in the clear in a single GPU (without SGX) and restrict access to this GPU using the same access control strategies. Using two servers and SGX doesn't buy you anything in this scenario.

* On optimizing CaffeScone:
Even if optimizing this baseline is not the main goal of your paper, it is still necessary to make the point that Goten is practical. The question is basically: if a company wants to spend resources to deploy private ML, should they implement an optimized Goten, or an optimized CaffeScone? If the optimized CaffeScone achieves similar performance, they'll be more likely to use that as it is much simpler to deploy and doesn't require non-colluding servers.

---

### Decision · Program_Chairs · 2019-12-19

**Decision:**

Reject

**Comment:**

This paper proposes a framework for privacy-preserving training of neural networks within a Trusted Execution Environment (TEE) such as Intel SGX. The reviewers found that this is a valuable research directions, but found that there were significant flaws in the experimental setup that need to be addressed. In particular, the paper does not run all the experiments in the same setup, which leads to the use of scaling factor in some cases. The reviewers found that this made it difficult to make sense of the results. The writing of this paper should be streamlined, along with the experiments before resubmission.